# Sensitivity of mixed-phase moderately deep convective clouds to parameterisations of ice formation - An ensemble perspective

Annette K. Miltenberger[1] and Paul R. Field[2,3]

[1]Institute for Atmospheric Physics, Johannes Gutenberg-University, Mainz, Germany
[2]Institute of Climate and Atmospheric Science, School of Earth and Environment, University of Leeds, United Kingdom
[3]Met Office, Exeter, United Kingdom

**Correspondence:** Annette K. Miltenberger (amiltenb@uni-mainz.de)

**Abstract.** The formation of ice in clouds is an important processes in mixed-phase and ice-phase clouds. Yet, the representation of ice formation in numerical models is highly uncertain. In the last decade several new parameterisations for heterogeneous freezing have been proposed. It is so far unclear what the effect of choosing one parameterisation over another is in the context of numerical weather prediction. We conducted high-resolution simulations ($\Delta x = 250\,\mathrm{m}$) of moderately deep convective clouds (cloud top $\sim -18^\circ\mathrm{C}$) over the southwestern UK using several formulations of ice formation and compare the resulting changes in cloud field properties to the spread of an initial condition ensemble for the same case.

The strongest impact of altering the ice formation representation is found in the hydrometeor number concentration and mass mixing ratio profiles. While changes in accumulated precipitation are around $10\,\%$, high precipitation rates ($95^{\mathrm{th}}$ percentile) vary by $20\,\%$. Using different ice formation representations changes the outgoing short-wave radiation by about $2.9\,\mathrm{W\,m^{-2}}$ averaged over daylight hours. The choice of a particular representation for ice formation has always a smaller impact then omitting heterogeneous ice formation completely. Excluding the representation of the Hallett-Mossop process or altering the heterogeneous freezing parameterisation has an impact of similar magnitude on most cloud macro- and microphysical variables with the exception of the frozen hydrometeor mass mixing ratios and number concentrations.

A comparison to the spread of cloud properties in a 10-member high-resolution initial condition ensemble shows that the sensitivity of hydrometeor profiles to the formulation of ice formation processes is larger than sensitivity to initial conditions. In particular, excluding the Hallet-Mossop representation results in profiles clearly different from any in the ensemble. In contrast, the ensemble spread clearly exceeds the changes introduced by using different ice formation representations in accumulated precipitation, precipitation rates, condensed water path, cloud fraction and outgoing radiation fluxes.

# 1 Introduction

Clouds consisting of a mixture of liquid and solid particles (mixed-phase) clouds play an important role for weather and climate at all latitudes. For example, observational data suggest that a significant fraction of surface precipitation form in mixed-phase clouds (e.g. Field and Heymsfield, 2015). It has also been demonstrated that the representation of cloud glaciation in global climate models has a substantial impact on the simulated mean climate state (e.g. McCoy et al., 2016). Despite this importance of mixed-phase clouds, for predicting weather and climate, the physical understanding of the underlying processes, most importantly ice formation, is very limited. Not surprisingly the representation of mixed-phase is one key source of uncertainty in weather and climate models (e.g. Korolev et al., 2017).

The formation of ice particles in the atmosphere has received particular attention over the last decades. Although the underlying physics of ice nucleation are still not understood, data from laboratory and field measurements has been used to suggest a number of new parameterisations that relate the aerosol population and environmental temperature to the number of nucleating ice crystals (e.g. DeMott et al., 2010, 2015; Niemand et al., 2012; Atkinson et al., 2013; Wilson et al., 2015). These new formulations gradually replace older formulations used in numerical weather prediction models (e.g. Cooper, 1986; Meyers et al., 1992). While it has been demonstrated that more sophisticated formulations of heterogeneous freezing, in particular its dependency on the aerosol population, is beneficial for predicting certain cloud types (e.g. Klein et al., 2009; Vergara-Temprado et al., 2018), it is not clear what the impact of choosing one parameterisation over another parameterisation is. A recent publication by Hawker et al. (2020) suggests that the increase of ice nucleating particle number concentration per unit decrease of temperature, i.e. the slope of the parameterisation, plays a key role in determining the impact of a specific parameterisation on the simulated tropical deep convective cloud field.

In addition to heterogeneous and homogeneous freezing of solution droplets, new ice particles can also be formed by so-called secondary ice formation processes, of which the Hallett-Mossop process is the most well known (e.g. Field et al., 2017). Although secondary ice formation seems to be crucial to explain observed ice crystal number concentration in many clouds, its representation in numerical models is highly uncertain and its importance for determining cloud properties is still debated (e.g. Field et al., 2017). Formulations for processes other than the Hallet-Mossop processes have only become available recently (e.g. Sullivan et al., 2018).

Here, we investigate the impact of using different heterogeneous freezing parameterisation and including a representation of the Hallett-Mossop process on the simulated evolution of moderately deep convective clouds (cloud top temperature around $-18\,°\mathrm{C}$) over the United Kingdom. Thereby we expand the study by Hawker et al. (2020) to a different cloud regime.

The standard approach to estimate the impact of altered cloud microphysical parameterisations is to conduct sensitivity experiments. The differences between the various experiments are interpreted as the impact of the parameterisation change. To assess the importance of the identified sensitivity in the context of model development and improvements for numerical weather prediction, it is, however, vital to compare the sensitivity to changes in one parameterisation to the uncertainty of the prediction due to other deficiencies in the model formulation and the overall predictability of the considered case. The latter is in particular important for convective situations with a small intrinsic predictability, as in these conditions any small

perturbation may rapidly amplify (e.g. Hohenegger and Schär, 2007; Dey et al., 2014). The relevance of taking into account the predictability of different situations for assessing the sensitivity to parameterisation changes is gaining increasing attention (e.g. Wang et al., 2012; Posselt et al., 2019). Quantifying the relative importance of initial condition uncertainty and uncertainty due to the model formulation is important for identifying priorities in future model development and justifying investment in more complex model formulations for operational weather forecasting centres. To address this issue, we place the sensitivity experiments in the context of a high-resolution initial condition ensemble.

The two key research questions addressed in the paper are:

- How sensitive are mixed-phase convective clouds with cloud top temperatures around $-18\,^{\circ}\mathrm{C}$ to the parameterisation of ice formation (heterogeneous freezing and Hallett-Mossop process)?

- How does the sensitivity to different descriptions of ice formation compare to typical initial condition uncertainty for day-1 forecasts?

The paper starts with a short introduction to the investigated case and the model framework used for the simulations (section 2). The results from the sensitivity experiments are presented in section 3 and place them in the context of the ensemble simulations in section 4. Finally, the key findings are summarised and discussed in section 5.

## 2 Model and data

Simulations were conducted for the $3^{\mathrm{rd}}$ August 2013 case from the COPE campaign Blyth et al. (2015); Leon et al. (2016); Miltenberger et al. (2018a). The campaign took place over the Southwestern Peninsula of the British Isles and probed convective clouds forming along converging sea-breeze fronts. We use the Unified Model vn10.3 with the Cloud Aerosol Interacting Microphysics Module (CASIM). The model set-up is identical to that described in Miltenberger et al. (2018a, b), key points of the set-up are summarised here and more detailed information is included in Appendix A. A regional nest with a grid spacing of $1\,\mathrm{km}$ resolution is nested in the global simulations, which in turn drives a second nest with a grid spacing of $250\,\mathrm{m}$. Only data from the innermost nest is used here. The initial and lateral boundary conditions for the $1\,\mathrm{km}$ nest are derived from the operational control run and nine members of the global operational ensemble forecast from the Met Office (MOGREPS, Bowler et al. (2008)), which represent the anticipated spread of moisture and moist energy convergence over the region of interest (see also Miltenberger et al. (2018b)). The aerosol environment is represented by using a constant profile for initial and boundary conditions, which has been derived from aircraft observation ("standard" aerosol scenario in Miltenberger et al. (2018a)), and by allowing for advection of aerosols in the domain. The aerosol profile is specified in terms of aerosol mass and number concentration of three soluble and one insoluble aerosol modes. The cloud microphysics are represented by the CASIM scheme, which is a two-moment scheme with five hydrometeor categories. In the configuration used here, aerosol properties influence cloud droplet and ice crystal formation, but the cloud microphysical processes do not alter the aerosol properties ("passive" mode in Miltenberger et al. (2018a)).

Substantial parametric and systematic structural uncertainty resides in the model representation of cloud microphysical pro-

cesses, in particular with regard to ice formation processes. Several heterogeneous freezing parameterisations, which differ in the used parameters and the form of the temperature dependence of ice formation (e.g. Hawker et al., 2020), have been suggested over the last decade. In order to investigate the implications of choosing specific schemes for numerical weather prediction, a set of new simulations has been conducted: seven simulations with different heterogeneous freezing parameterisation ("FSENS"), one simulation omitting the parameterisation of the Hallett-Mossop process ("NoHM"), and one omitting all ice-phase processes ("WARM"). The Hallett-Mossop process describes the generation of so-called secondary ice crystals during riming of snow and graupel, in the model a fixed number of additional small ice crystals are generated per gram of accumulated rime. This number is temperature dependent with a maximum at an ambient temperature of $-5\,°C$ and decay to zero at temperatures of $-3\,°C$ and $-8\,°C$, respectively. Initial and lateral boundary conditions for these sensitivity experiments are derived from the operational global control run. For the FSENS experiments we used the heterogeneous freezing parameterisations by Meyers et al. (1992) (M92), Atkinson et al. (2013) (A13), DeMott et al. (2010) (DM10), DeMott et al. (2015) (DM15), and Niemand et al. (2012) (N12). The DM10 parameterisation is used in the "NoHM" simulation. The Meyers et al. (1992) parameterisation predicts the number of ice crystals based only on the temperature at the location of interest and thereby is the conceptually "simplest" parameterisation used here. The DeMott et al. (2010) and DeMott et al. (2015) parameterisation take in addition to the temperature the aerosol number concentration of particles with diameters larger than $0.5\,\mu m$ into account. This parameterisation thereby provides an empirical link to the number of INP particles that are expected to be present given the aerosol concentration at the location of interest. Finally, the parameterisations by Niemand et al. (2012) and Atkinson et al. (2013) are based on a temperature-dependent parameterisation of the number of active sites per unit surface area of dust aerosol. This number density of actives sites is then convolved with the total surface aerosol of dust aerosol to obtain again the number of INPs. The different representations of ice nucleation thereby reflect different levels of detail in the representation of heterogeneous ice formation.

When the number of INPs as a function of temperature is considered for a fixed aerosol concentration Hawker et al. (2020) showed that the important controlling factors are the slope of the temperature dependence, i.e. $\frac{\mathrm{dINP}}{\mathrm{dT}}$, and the temperature at which the INP number concentration reaches the dust number concentration, i.e. the maximum possible value. The combined effect of these differences results in the Meyers et al. (1992) parameterisation having the highest INP concentration at temperatures and the Atkinson et al. (2013) parameterisation having the lowest INP concentration warmer than about $-18°C$. At temperatures colder than about $-18°C$ the Atkinson et al. (2013) parameterisation has the largest and the DeMott et al. (2010) parameterisation the lowest INP number concentrations. In addition to the sensitivity experiments with different heterogeneous freezing parameterisations, two simulations with pre-factors of $10$ and $0.1$ for the DM10 parameterisation are included, which represent high- and low-INP regimes induced by changes in aerosol concentration. The simulation with the DM10 parameterisation is identical to the the "control" simulation in Miltenberger et al. (2018b) and is referred to as "baseline" simulation in the following.

The baseline simulation with the DM10 parameterisation has been compared to observational data in Miltenberger et al. (2018a). In that paper, we found a good agreement of the thermodynamic structure of the simulated atmosphere and the 2-hourly radiosondes from Davidstow: In particular, a stable layer at the height of about $5-6\,\mathrm{km}$, which determines the

maximum cloud top depth, is found in the model at approximately the same altitude ($\pm 50\,\mathrm{hPa}$) and with the comparable stability. Also the altitude of the $0\,^{\circ}\mathrm{C}$ isotherm and the lifting condensation level were found to be deviate by less than $250\,\mathrm{m}$ from the observed position. In the sensitivity experiments discussed here no systematic difference in the thermodynamic structure is found (not shown). Additionally, Miltenberger et al. (2018a) compared the vertical velocity dependence of the cloud droplet number concentration at cloud base in the model to in-situ observations taken with the Bae-146. Qualitatively the dependence of the cloud droplet number concentration on the cloud-base updraft velocity is very similar in model and observations, but absolute values are overestimated by about $30\,\%$ in the model. A qualitative comparison of the variation of cloud droplet number concentration with altitude indicates a stronger decrease of cloud droplet number concentration with altitude in the observations. This is most likely due to the aerosol treatment in the "passive" aerosol mode of CASIM used also here, as simulations with a fully-interactive aerosol version of CASIM showed improvements. Comparison to ice-phase particle concentrations is difficult due to the specific sampling strategy in rising turrets, which can suffer from bias (Field and Furtado, 2016) and is not easily mimicked from the modelling data. To get further insight into the representation of the cloud field as a whole, 3D radar data was utilised in Miltenberger et al. (2018a): The mean and maximum cloud top height, measured by the maximum altitude of the $18\,\mathrm{dBZ}$ contour, compared well between model and radar (differences $< 500\,\mathrm{m}$). As there is little change in the mean or maximum cloud top height in the sensitivity experiment, determined by the maximum altitude of cloud condensate (radar reflectivity data is not available for the sensitivity experiments) this comparison is not expected to deteriorate. Only the WARM simulation is showing systematically lower mean cloud top heights, which will likely result in a some what less favourable comparison to radar observations. In addition, the timeseries of domain-average surface precipitation in the simulations was compared to the radar-derived surface rainfall rates (Radarnet IV rainfall retrieval, Harrison et al. (2009)). The baseline model simulation was found to underestimate surface precipitation in the morning hours and the early evening by about $80\,\%$, but the temporal structure and the main precipitation period (precipitation rate within $10\,\%$) are both captured well. The same comparison including the sensitivity experiments analysed in this paper is shown in SI Fig. S1. Again the sensitivity experiments show neither a systematically enhanced nor deteriorated performance. Only the reduced precipitation rates in the WARM experiments seem less comparable with the observations. In summary, we have demonstrated that the baseline simulation successfully captures many features of the observed cloud and precipitation evolution, the thermodynamic conditions and cloud microphysical parameters. This conclusion holds also for all sensitivity experiments here, with the exception of the WARM case. A more detailed comparison with in-situ observations of ice phase or the 3D radar reflectivity structure would be interesting , but is beyond the scope of the present paper. Accordingly, the set-up provides a meaningful framework for the sensitivity analysis presented here.

## 3  Sensitivity of cloud field properties to representation of ice formation

Varying the representation of primary and/or secondary ice formation has a direct impact on the number of ice crystals produced at a specific temperature, and hence ice crystal number concentrations (ICNC) vary between the different experiments. Despite a multitude of other processes altering ICNC in a complex cloud field, systematic variations in the average ICNC

profile appear in the different experiments (Fig. 1 c). The profiles used here are average in-cloud profiles over the time period 10 UTC to 19 UTC. Differences are largest towards cloud top, with a spread of about one order of magnitude at 5 km altitude. Cloud bases are located roughly at 1 km altitude, cloud tops are located at $5.5 - 6$ km altitude and the $0\,°C$ level is found at around 2.6 km altitude (Miltenberger et al., 2018a). In the altitude range, where the Hallet-Mossop processes is active (i.e. $3 - 4$ km altitude corresponding to roughly $-3$ to $-8\,°C$), ICNC concentrations vary by about a factor 2 between the FSENS experiments, while ICNC concentrations in the NoHM run are about 1.5 orders of magnitude smaller than in any FSENS experiment. Despite this clear signal of the Hallett-Mossop process in the $3 - 4$ km altitude range, ICNC towards cloud top reaches similar values as in the FSENS experiments.

The differences in ICNC can impact the occurrence of other hydrometeor species via various cloud microphysical processes (Fig. 1): Snow crystal concentrations vary by up to a factor 2 between the different FSENS experiments and it is by a factor 5 lower in the NoHM experiment. In contrast to the signal in ICNC, the imprint of the Hallett-Mossop processes is consistent throughout the cloud layer. Interestingly, the variation in graupel number concentration is largest of all frozen hydrometeor types. Again the NoHM simulation displays the lowest number concentration. Altering the representation of ice formation also impacts the number concentration of liquid hydrometeors, particularly in the upper cloud parts: While the cloud droplet number concentration (CDNC) in the WARM simulation is almost constant with altitude, CDNC is significantly reduced in the FSENS and NoHM experiments above about 3 km. This is likely a consequence of freezing and collection by ice, snow and graupel particles. Interestingly, FSENS experiments with a high ICNC above 5 km have a low CDNC and vice-versa, implying a major impact of cloud droplet freezing . Variations in rain number concentrations are somewhat smaller than in CDNC. The profiles from the NoHM experiment feature roughly in the middle of the FSENS experiments for both cloud droplet and rain drop number concentration, i.e. the main impact of the Hallet-Mossop process is limited to frozen hydrometeor species in our simulations. If instead of the mean number concentration the $95^{th}$ percentile is considered, the general behaviour is very similar to that just discussed for the mean profiles (SI Fig. S2). The one outstanding differences is a much larger ice crystal number concentration in the simulation with enhanced INP concentrations ("HighDM"). This suggests that while higher INP concentrations result in an enhanced ice crystal formation, as is to be expected, the impact on mean ice crystal number concentration is much smaller due to the depletion of ice crystals by other microphysical processes, such as for example conversion to snow or graupel.

The average profiles of hydrometeor mass mixing ratios essentially mimic the sensitivities just discussed for the hydrometeor number concentrations (Fig. 2). Ice, snow and graupel mass mixing ratios are consistently lower in the NoHM experiment than in all other experiments. Differences in ice, cloud droplet and rain drop mass mixing ratios occur mainly in the upper part of the clouds (above $\sim 3.5$ km), while variation in snow (graupel) mass mixing ratio are small (large) throughout the entire cloud layer.

Different representations of ice formation clearly impact the cloud microphysical structure of the moderately deep convective clouds from COPE. We now investigate how these changes impact larger-scale features of the cloud field, such as the structure of the surface precipitation field, accumulated precipitation and top-of-the-atmosphere radiation fluxes. The instantaneous precipitation rate at 14 UTC, e.g. close to the time of most intense precipitation, is shown in Fig. 3. In all simulations a line of

organised convection is extending roughly along the center line of the peninsula (i.e. WSW-ENE). Overall, the structure of the precipitation field is similar in terms of area covered, number of cells, and peak intensity of the cells. The variability between the different sensitivity experiments is comparable to the difference between different members of the initial condition ensemble using the DeMott et al. (2010) parameterisation (SI Fig. S3). For a more quantitative analysis, we consider accumulated surface precipitation and the precipitation rate distribution. Accumulated surface precipitation varies by about 8 % between FSENS experiments (Fig. 4 a). While omitting secondary ice formation leads to an increase in accumulated precipitation of about $\sim$ 6 % relative to the baseline simulation, omitting all ice formation results in a reduction of accumulated precipitation by about $\sim$ 21 %. It is not straightforward to understand the changes in accumulated precipitation from the differences in the cloud microphysical composition of the clouds. Therefore, we choose to investigate the cloud condensate budget as suggested for example by Khain (2009) and Miltenberger et al. (2018a). In this analysis framework, the changes in cloud condensate in the domain are analysed: Cloud condensate can be generated in regions of lifting that can then be converted to surface precipitation, or be depleted by evaporation and sublimation (condensate loss). In addition, the total cloud condensate content in the domain can be influenced by advective fluxes, but Miltenberger et al. (2018a) show that for the present case this has a negligible impact. Condensate generation is strongly controlled by thermodynamics and dynamics, i.e. the amount of lifting in the domain and the vertical temperature structure, and to a smaller degree to deposition growth in mixed- and phase-clouds (the UM employs a saturation adjustment scheme). The partitioning between condensate forming precipitation and condensate loss are much more strongly influenced by cloud microphysical processes. To investigate changes in precipitation between simulations, the changes in condensate generation and condensate loss can provide insight into the driving processes. Differences in accumulated condensate generation G and condensate loss L are calculated relative to the baseline simulation, i.e. using DM10. In the scatterplot of $\Delta$G against $\Delta$L FSENS and NoHM experiments cluster on the one-to-one line (Fig. 5 a). Simulations falling exactly on the one-to-one line in this diagram have the same surface precipitation, as change in condensate generation and condensate loss compensate each other. The red (blue) dashed lines in Fig. 5 a indicate the distance away from the one-to-one line that correspond to a decrease (increase) of accumulated precipitation by 0.5 %, 5 %, and 10 %. Except for the WARM simulation, changes in accumulated precipitation are smaller than about 5 % as already shown in Fig. 4 a. Relative changes in G and L are $\leq$ 2 % for FSENS experiments. In the NoHM experiments changes to G and L are larger ($\sim$ 4 %), but balance each other resulting in a small net change in accumulated precipitation. Combined with the much larger changes in the cloud microphysical structure, this implies that changes in precipitation formation via a specific cloud microphysical pathways are compensated to a large degree by changes in other pathways resulting in an overall similar integrated precipitation production. The only experiment displaying a different behaviour is the WARM experiment: While condensate generation decreases by $\sim$ 5 %, condensate loss only decreases by $\sim$ 0.1 %. The reduction in accumulated precipitation compared to the baseline simulation is hence the result of much less condensate being produced in the WARM experiment. If assuming the vertical displacement of parcels does not change between simulations and any produced supersaturation is depleted by condensate formation, this is consistent with the lower saturation vapour pressure over ice than over water. However, without supporting evidence this remains a hypothesis. Further, a negative $\Delta$G and no change in $\Delta$L implies that the precipitation efficiency in the WARM experiment is larger than in any experiment with ice microphysics. Precipitation efficiency is defined here as the

ratio of time- and domain-integrated precipitation rate to condensation and deposition rate. This response is contrary to what has been reported for isolated orographic clouds (e.g. Barstad et al., 2007; Miltenberger, 2014) and the larger precipitation efficiency for more rapidly glaciating clouds in high-INP environments found in global climate model simulations (e.g. Lohmann and Hoose, 2009). However, a reduction in precipitation efficiency with an increased cloud glaciation has been also found by Levin et al. (2005) for convective clouds in the Mediterranean.

Similar to the accumulated precipitation, the precipitation rate distribution displays only a weak sensitivity to the parameterisation used for the representation of primary ice formation (Fig. 4 b). Again, the only experiment with a substantially different behaviour is the WARM experiment, which displays a shift towards more intense precipitation: High precipitation rates ($\geq 20\,\mathrm{mm\,h^{-1}}$) are more frequent, while medium rain rates between $1\,\mathrm{mm\,h^{-1}}$ and $10\,\mathrm{mm\,h^{-1}}$ are about $10\,\%$ less frequently. Very high precipitation rates, i.e. the $95^{\mathrm{th}}$ and $99^{\mathrm{th}}$ percentile, display the largest changes. The $95^{\mathrm{th}}$ percentile varies by about $20\,\%$ between FSENS experiments and increases by $50\,\%$ in the WARM experiment compared to the mean of the FSENS experiments (SI Fig. 3).

The condensed water path and the cloud fraction are other important properties of the cloud field. The difference in the condensed water path between FSENS and NoHM experiments is $29\,\%$ of the water path in the baseline simulation $((\mathrm{CWP(t)_{max}} - \mathrm{CWP(t)_{min}})/\mathrm{CWP(t)_{baseline}})$ in the late afternoon ($\sim 15 - 17$ UTC), but smaller values prevail at other times resulting in an average maximum spread between FSENS and NoHM experiments of $14\,\%$ (Fig. 6 a). In the WARM experiment the condensed water path is lower than in any other experiment throughout most of the afternoon (maximum: $41\,\%$, mean: $16\,\%$ reduction compared to the baseline experiment). This is consistent with the smaller condensate generation and enhanced precipitation efficiency diagnosed for this experiment. Changes in cloud fraction between the different experiments amount at maximum to $20\,\%$ of the value in the baseline experiment (Fig. 6 b). Cloud fraction is defined as the areal fraction of the domain with column-integrated condensed water path larger than $1\,\mathrm{g\,m^{-2}}$. Again, the maximum differences occur in the late afternoon hours. Averaged over the entire time-period, the changes are much smaller ($7\,\%$).

Finally, we also consider the sensitivity of outgoing shortwave and longwave radiation (Fig. 7). The maximum domain mean difference between any two FSENS/NoHM experiments is about $6\,\mathrm{W\,m^{-2}}$ for the shortwave component and $0.5\,\mathrm{W\,m^{-2}}$ for the longwave component. The average over the considered time-period amounts to $2.9\,\mathrm{W\,m^{-2}}$ ($0.27\,\mathrm{W\,m^{-2}}$) for the shortwave (longwave) component. Similar to the other cloud field characteristics discussed so-far the largest change occurs in the WARM experiment with a maximum (average) increase of $15\,\mathrm{W\,m^{-2}}$ ($5.7\,\mathrm{W\,m^{-2}}$) in the shortwave component and a maximum (average) decrease of $1.4\,\mathrm{W\,m^{-2}}$ ($0.5\,\mathrm{W\,m^{-2}}$) in the longwave component.

Considering the temporal evolution of most cloud properties, i.e. domain-integrated precipitation (not shown), condensed water path (Fig. 6 b) and top-of-the-atmosphere outgoing radiation (Fig. 7), the consistency in the evolution between different experiments is noteworthy, which strongly suggests that the COPE clouds are strongly dynamically forced with little leeway for cloud microphysics to change the overall characteristics of the cloud field.

Overall the sensitivity to the representation of ice formation found here for moderately deep convective clouds (cloud top $\sim -18\,^{\circ}\mathrm{C}$) is smaller than reported for tropical deep convective clouds (e.g. Hawker et al., 2020). Hawker et al. (2020) find differences of up to $21\,\mathrm{W\,m^{-2}}$ in the total outgoing radiation in a set of simulations comparable to our FSENS experiments.

The majority of the signal reported in Hawker et al. (2020) is due to changes in anvil properties. This likely explains the smaller signal in our simulations, as the investigated convective clouds are shallower and do not produce spatially extensive anvil clouds. In particular, in the context of numerical weather prediction, but also for deriving observational constraints on the cloud microphysical parameterisations, it is important to understand how these sensitivities compare to uncertainty in modelled cloud field properties due to other factors such as initial condition uncertainty or uncertainties in the formulation of other model components. To provide some context for the sensitivities discussed here, we compare them in the next section with the spread of a 10-member high-resolution initial condition ensemble.

## 4    Comparison to sensitivity to initial condition perturbations

The representation of ice formation has a fairly strong impact on the cloud microphysical properties of clouds and can induce changes of between $5 - 20 \, \%$ in cloud field properties, such as accumulated precipitation, cloud fraction, and outgoing radiation fluxes (see section 3, summarised in Table 1 in terms of the relative spread). In order to judge the significance of these variations, it is necessary to put them into the context of other uncertainty sources for the modelled cloud properties. As forecasts of convective situations often have a low intrinsic predictability (e.g. Hohenegger and Schär, 2007), it is particularly interesting to use ensemble simulations with perturbed initial conditions as context for sensitivity experiments regarding the model formulation. Here, we use high-resolution ensemble simulations for the COPE case, which were already used by Miltenberger et al. (2018b) to provide context for sensitivity experiments regarding the background aerosol concentration. We focus here on comparing the spread of variables between the ensemble members to the spread between different sensitivity runs. The spread from ensemble runs is indicated in all figures by the grey shaded area.

Altering the representation of ice formation impacts the hydrometeor number, particularly that of ice crystals (ICNC) and cloud droplets (CDNC) in the upper layers (above $\gtrsim 4.5 \, \mathrm{km}$ and $\gtrsim 3 \, \mathrm{km}$, respectively). These changes are much larger than the maximum spread in mean hydrometeor number profiles from the ensemble (Fig. 1 a and c). In contrast, the sensitivity of rain and graupel number densities to different ice formation representations (FSENS) is comparable to the sensitivity of the modelled clouds to perturbations in the initial conditions (Fig. 1 b and e). For snow, changes in number concentration across FSENS experiments are clearly smaller than the impact of perturbed initial conditions. Regarding the impact of secondary ice formation, here in the form of the Hallett-Mossop process, it is intriguing to note that the NoHM experiments yield mean hydrometeor profiles that are clearly outside of the ensemble spread for all frozen hydrometeor species.

In general the picture is very similar when hydrometeor mass mixing ratios are considered instead of their number densities (Fig. 2). The sensitivity to the ice formation representation is larger than the initial condition ensemble spread for upper-level cloud droplet and ice crystal content as well as additionally the rain water content. The NoHM experiments again have profiles outside the range from the ensemble for all hydrometeor species, but with a smaller separation from the ensemble for snow and graupel compared to the number concentration profiles (Fig. 2 d and e). Overall it appears that the sensitivity to ice formation representation is larger than that to initial conditions perturbations even for the mean hydrometeor profiles.

If instead of the cloud microphysical structure the properties of the cloud field are considered the picture changes: Considering,

for example, the accumulated surface precipitation the differences between FSENS and NoHM experiments is only very small if compared to the spread between members in the initial condition ensemble (Fig. 4 a). The ratio between the spread from the sensitivity experiments (FSENS & NoHM) to the spread of the ensemble is roughly 0.2. Even the difference between the baseline and the WARM experiments is much smaller than the ensemble spread. Not surprisingly, also the differences in the condensate budget are much larger across the initial condition ensemble compared to the sensitivity experiments (Fig. 5 b). However, if precipitation efficiency is considered the variability across ensemble members ($0.176 - 0.256$) and sensitivity experiments ($0.180 - 0.230$) is again very similar (not shown). This suggests that the dominance of initial condition uncertainty for the accumulated precipitation is due to the strong control of larger-scale moisture and moist static energy convergence. For the conversion of this condensate to precipitation, however, the representation of cloud microphysical processes is at least as important as the larger-scale meteorological conditions. In the investigated case, variability in condensate generation clearly exceeds the impact of the variability in precipitation efficiency and hence the former dominates the predicted spread of accumulated precipitation.

Similar to accumulated precipitation, also for condensed water path, cloud fraction as well as short- and long-wave outgoing radiation the spread between ensemble members is much larger than their sensitivity to a particular representation of ice formation (Fig. 6 & 7). The relative spread between various sensitivity experiments and ensemble members is summarised in Table 1.

Our analysis suggests that, at least for the investigated case forecast uncertainty is dominated by initial condition uncertainty for all cloud field variables, while uncertainty intrinsic to the representation of ice formation (reflected by parameterisation choice) place only for the detailed cloud microphysical structure a dominant role.

## 5 Discussion and Conclusions

We investigate the sensitivity of model predictions of a moderately deep convective cloud field to altered representations of ice formation (different heterogeneous freezing parameterisations, representation of Hallet-Mossop process) and to initial condition uncertainty for lead times of up to $19\,\mathrm{h}$. The investigated case was selected from those observed in the COPE campaign (e.g. Leon et al., 2016). The case was already investigated in Miltenberger et al. (2018a, b) with a focus on aerosol-cloud interactions.

Altering the ice formation representation impacts the cloud microphysical structure, in particular the cloud droplet, ice crystal and graupel mass mixing ratio and number concentration, as well as cloud field properties such as surface precipitation, cloud fraction and outgoing short- and long-wave radiation. Accumulated surface precipitation varies by about $8\,\%$ ($21\,\%$) and mean cloud fraction by about $7\,\%$ ($7\,\%$) across experiments with different descriptions of ice formation (only warm-phase cloud microphysics). Average outgoing short-wave radiation changes by $2.9\,\mathrm{W\,m^{-2}}$ ($2.9\,\mathrm{W\,m^{-2}}$) and outgoing long-wave radiation by $1.4\,\mathrm{W\,m^{-2}}$ ($0.5\,\mathrm{W\,m^{-2}}$) in the respective set of experiments. The sensitivity to the representation of ice formation in our case is smaller than the sensitivity found by Hawker et al. (2020) for tropical deep convective clouds. In Hawker et al. (2020), the anvils of convective clouds contributed significantly to the overall changes in cloud fraction and outgoing radiation com-

ponents. In contrast, to their case cloud in our case only reach up to a stable layer in the mid-troposphere (Miltenberger et al., 2018a) and no anvils are present. This likely explains the smaller sensitivity to ice formation representation.

The importance of the observed sensitivity to ice formation representation for numerical weather forecasting depends on how
it compares to other sources of uncertainty for predicting the cloud field evolution, including initial condition uncertainty and parametric or systematic uncertainty in other model components. In the present work, we use a high-resolution initial condition ensemble to provide context for the sensitivity experiments. From comparing the ensemble spread to the differences between sensitivity experiments it becomes clear that for bulk cloud field properties such as accumulated precipitation, cloud fraction and outgoing radiation initial condition uncertainty clearly exceeds the sensitivity to the formulation of ice forma-
tion. However, for the mean hydrometeor profiles, in particular cloud droplet, ice crystal and graupel mass mixing ratios and number concentration, initial condition uncertainty is less important than the choice in ice formation parameterisation. The impact of the Hallett-Mossop process is particularly evident as the mean profiles in simulations without a representation of the Hallett-Mossop processes are clearly outside of the ensemble spread. While this may indicate a significant role of secondary ice formation in this cloud type, the representation of secondary ice formation in clouds is itself highly uncertain and this
uncertainty has not been explored here. The large impact of initial and boundary conditions on the bulk cloud field properties derives from the strong control of moisture and moist static energy convergence on these. Combined with the clearly different cloud microphysical structure of the clouds, this implies that altering the chosen ice formation parameterisations impacts the pathway of precipitation formation, albeit with a small impact on the larger-scale cloud properties, i.e. suggesting the considered mixed-phase cloud systems maintains its large scale properties regardless of changes in the balance of the microphysical
pathways.

It would be interesting to compare the sensitivity to ice formation parameterisation with the impact of other parametric uncertainties in the model. In a previous study, we have investigated the sensitivity of the same case to alterations of the aerosol background concentration (factor 10 increase and decreases, respectively) (Miltenberger et al., 2018a, b). We found that the cloud field is also less sensitive to changes in aerosol conditions than to perturbations of initial conditions, at least if larger-
scale properties such as accumulated precipitation, cloud fraction and radiative fluxes are considered. In sum, this suggests that COPE-type clouds are strongly controlled by meteorological conditions with comparatively little leeway for cloud microphysics to modify cloud field properties.

Of course the question arises, whether this dominance of initial condition uncertainty is a special feature of the chosen case. To date only few studies combine an ensemble approach with sensitivity experiments (e.g. Seifert et al., 2012) and most of
these focus on idealised cases (e.g. Grabowski et al., 1999; Morrison, 2012; Wang et al., 2012; Posselt et al., 2019; Wellmann et al., 2019). Nevertheless, the overall findings are compatible with the present study, in that bulk properties such as radiative fluxes and accumulated precipitation, are strongly influenced by larger-scale meteorological conditions and to a lesser degree by perturbations to the cloud microphysical scheme, be it perturbations to the aerosol environment (e.g. Seifert et al., 2012; Grabowski et al., 1999; Morrison, 2012) or to the formulation of cloud microphysical processes (e.g. Wang et al., 2012; Posselt
et al., 2019; Wellmann et al., 2019). Recently, several studies ventured to systematically investigate the joint impact of multiple uncertain parameters in the cloud microphysics representation, although again these studies have been largely focussed

on idealised case (e.g. Johnson et al., 2015; Glassmeier et al., 2019). For idealised simulations of deep convection, Johnson et al. (2015) found a small impact of parameters in the immersion freezing parameterisation on accumulated precipitation compared to the impact of other parameters in the cloud microphysical parameterisation, such as collection efficiencies and aerosol number concentration, which is consistent with our COPE studies. In the absence of more studies systematically combining perturbations to aerosol conditions and / or cloud microphysical processes, we can only speculate about impact of such perturbations beyond initial condition uncertainty. The COPE case represents a convective situation with a fairly strong forcing from the converging sea-breeze fronts and convergence of moisture into the study area. In cases with a weaker dynamic forcing, such as unorganised convection, we expect a lesser impact of the initial condition uncertainty. However, whether uncertainty in the cloud microphysics representation and if in particular which cloud microphysical processes will dominate over initial condition uncertainty is difficult to assess a-priori. If the changes in the cloud microphysics representation lead to a systematic shift that is consistent in sign across a large range of conditions, that should become clearly visible if a large number of cases are considered. How large such a set of cases needs to be depends on the degree to which large-scale meteorological conditions are constrained as well as on the relative impact of these meteorological conditions and the model perturbations on the variables of interest (for an example concerning an assessment of this for aerosol perturbations in COPE-like scenarios see Miltenberger et al. (2018b)).

In summary, the simulations show that differences in ice formation parameterisation primarily impact the cloud microphysical structure with less impact on cloud field properties. Although broadly consistent with previous work, the study presented here has some shortcomings, which we plan to address in future work. Mainly it would be desirable to repeat the full ensemble simulations with the changes to the cloud microphysics representation, to investigate number of joint parameter perturbations, to test the sensitivity to the choice of the domain (e.g. White et al., 2018), and to repeat the analysis for different cases.

*Data availability.* Model data is stored on the tape archive provided by JASMIN (http://www.jasmin. ac.uk/) service. Data access to Met Office data via JASMIN is described at http://www.ceda.ac.uk/blog/access-to-the-met-office-mass-archive-on-jasmin-goes-live/. The data can be made accessible upon request to the authors.

## 6  Appendix A: Additional detail on the model set-up

The regional simulations used in this paper are run without a convection parameterisation and without a cloud scheme. Boundary layer processes are parameterised with the blended boundary layer scheme by Lock et al. (2015) and turbulence with a 3-D Smagorinsky-type turbulence scheme. Radiation fluxes are parameterised with the SOCRATES scheme (Suite of Community RAdiative Transfer codes, (Edwards and Slingo, 1996; Manners, 2017)). In addition, moisture conservation in the regional domain is enforced by using the scheme by Aranami et al. (2014, 2015).

Cloud microphysical processes are represented by the CASIM microphysics. The cloud particle population is represented by the mass and number concentration of five hydrometeor categories assuming gamma distributions. In addition, mass and

number concentration of four aerosol modes are computed based on the prescribed initial and lateral boundary conditions as well as advection inside the domain. Droplet activiation is parameterised following Abdul-Razzak et al. (1998); Abdul-Razzak

and Ghan (2000). Further represented processes include condensational growth by saturation adjustment scheme, freezing of rain drops Bigg (1953), homogeneous freezing of cloud droplets Jeffery and Austin (1997), Hallett-Mossop processes, vapour deposition, collision-coalescence processes between all hydrometeors, and gravitational settling of all hydrometeor categories except cloud droplets. For heterogeneous freezing of cloud droplets different parameterisations are used as detailed in section 2.

The initial and lateral boundary conditions for aerosols are based on aircraft observations from the COPE campaign. Different aerosol number and mass concentrations are prescribed in the planetary boundary layer and the free troposphere with a linear transition between the two values. The transition zone is centered at 1.15 km and has a depth of 500 m. Aerosol number and mass concentrations are constant with altitude in the boundary layer and the free troposphere. The values used for the boundary layer are given in table A1 below.

*Author contributions.*  All authors contributed to the development of the concepts and ideas presented in this paper. A. K. Miltenberger set up and run the model simulations. She also performed the model analysis and wrote the majority of the manuscript, along with input and comments from P. R. Field.

*Competing interests.*  The authors declare that they have no conflict of interest.

*Disclaimer.*  TEXT

*Acknowledgements.*  We thank Adrian Hill, Ben Shipway and Jonathan Wilkinson for the development of the CASIM module. We acknowledge use of the Monsoon/NEXCS system, a collaborative facility supplied under the Joint Weather and Climate Research Programme, a strategic partnership between the Met Office and the Natural Environment Research Council. Further, we acknowledge JASMIN storage facilities (doi : 10.1109/BigData.2013.6691556). The University of Leeds and Johannes Gutenberg University Mainz are acknowledged for providing funds for this study. We thank two anonymous reviewers for their comments on the manuscript.

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

**Table 1.** Relative spread, i.e. difference between maximum and minimum value in any simulation divided by the value in the base line simulation, for: mean cloud droplet number concentration (CDNC), ice crystal number concentration (ICNC), cloud mass mixing ration ($q_c$), frozen hydrometeor mass mixing ratio ($q_f$), accumulated surface precipitation (P), condensed water path (TWP), cloud fraction, and outgoing short-wave (OSR) as well as long-wave (OSR) radiation.

| Variable | ensemble (max-min) | FSENS (max-min) | FSENS, DM10 (max-min) | baseline - NoHM | baseline - WARM |
|---|---|---|---|---|---|
| CDNC ($>$ 4.5 km) | 1.40 | 2.98 | 0.671 | 0.814 | 7.84 |
| ICNC ($>$ 4.5 km) | 0.961 | 3.36 | 3.19 | 0.503 | - |
| ICNC ($<$ 4.5 km) | 0.439 | 1.29 | 0.969 | 0.876 | - |
| $q_c$ ($>$ 4.5 km) | 5.15 | 5.58 | 0.862 | 2.19 | 12.9 |
| $q_f$ ($>$ 4.5 km) | 2.72 | 0.601 | 0.373 | 0.595 | - |
| $q_f$ ($<$ 4.5 km) | 0.520 | 0.196 | 0.120 | 0.472 | - |
| P | 0.587 | 0.0798 | 0.0254 | 0.0594 | 0.209 |
| TWP | 0.941 | 0.140 | 0.0890 | 0.0775 | 0.159 |
| cloud fraction | 1.32 | 0.0643 | 0.0404 | 0.0246 | 0.0375 |
| OSR | 0.154 | 0.0190 | 0.00948 | 0.0119 | 0.0321 |
| OLR | | 0.0141 | 0.00102 | 0.000523 | 0.00188 |

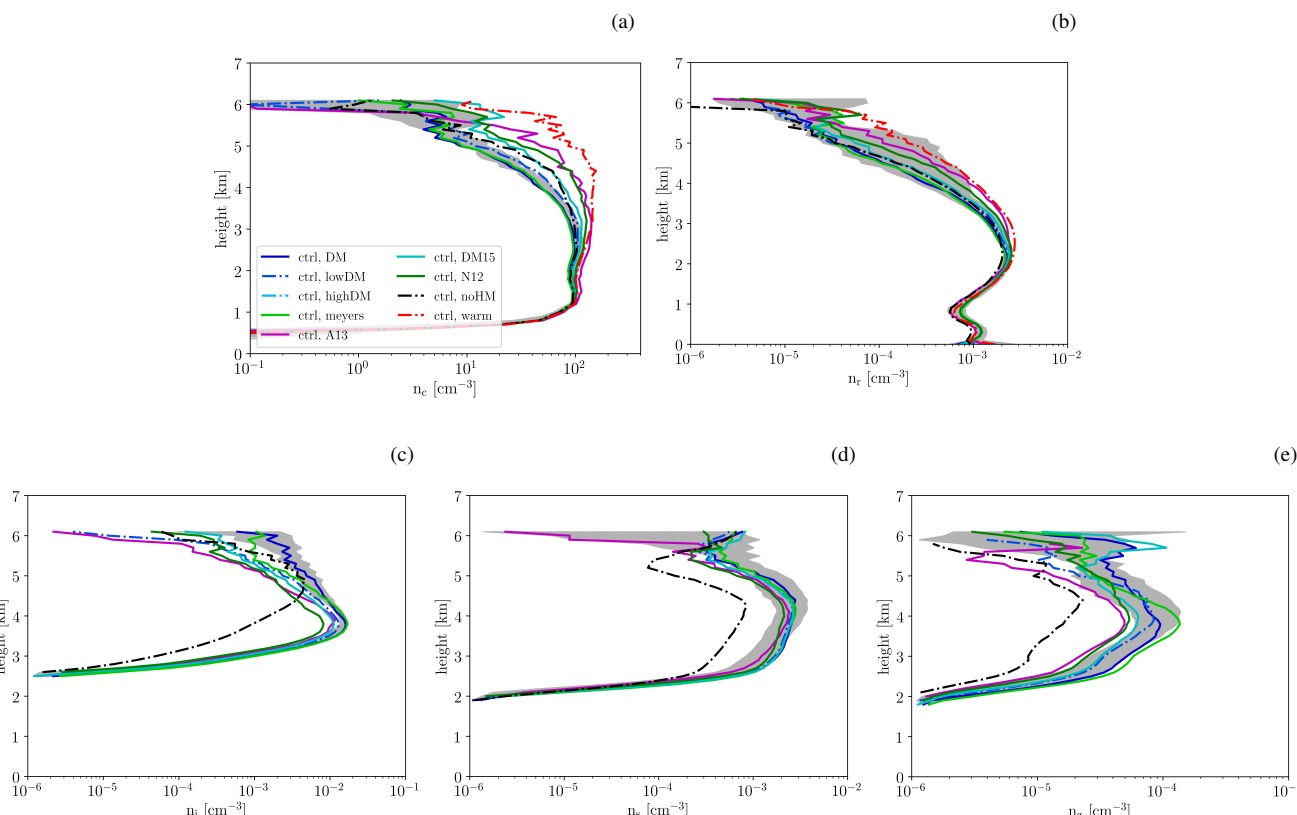

**Figure 1.** Average profiles of in-cloud number concentrations of (a) cloud droplets, (b) rain drops, (c) ice crystals, (d) snow and (e) graupel. Different coloured lines show the profiles from simulations with different heterogeneous freezing parameterisations, different INP number concentrations, without a parameterisation of the Hallet-Mossop process and with warm cloud microphysics only (colours according to legend). The grey shading shows the spread of the average profiles in the 10-member high-resolution ensemble with the DeMott et al. (2010) heterogeneous freezing parameterisation and a representation of the Hallett-Mossop process. The $0\,^{\circ}$C level is located at about $2.6\,$km altitude.

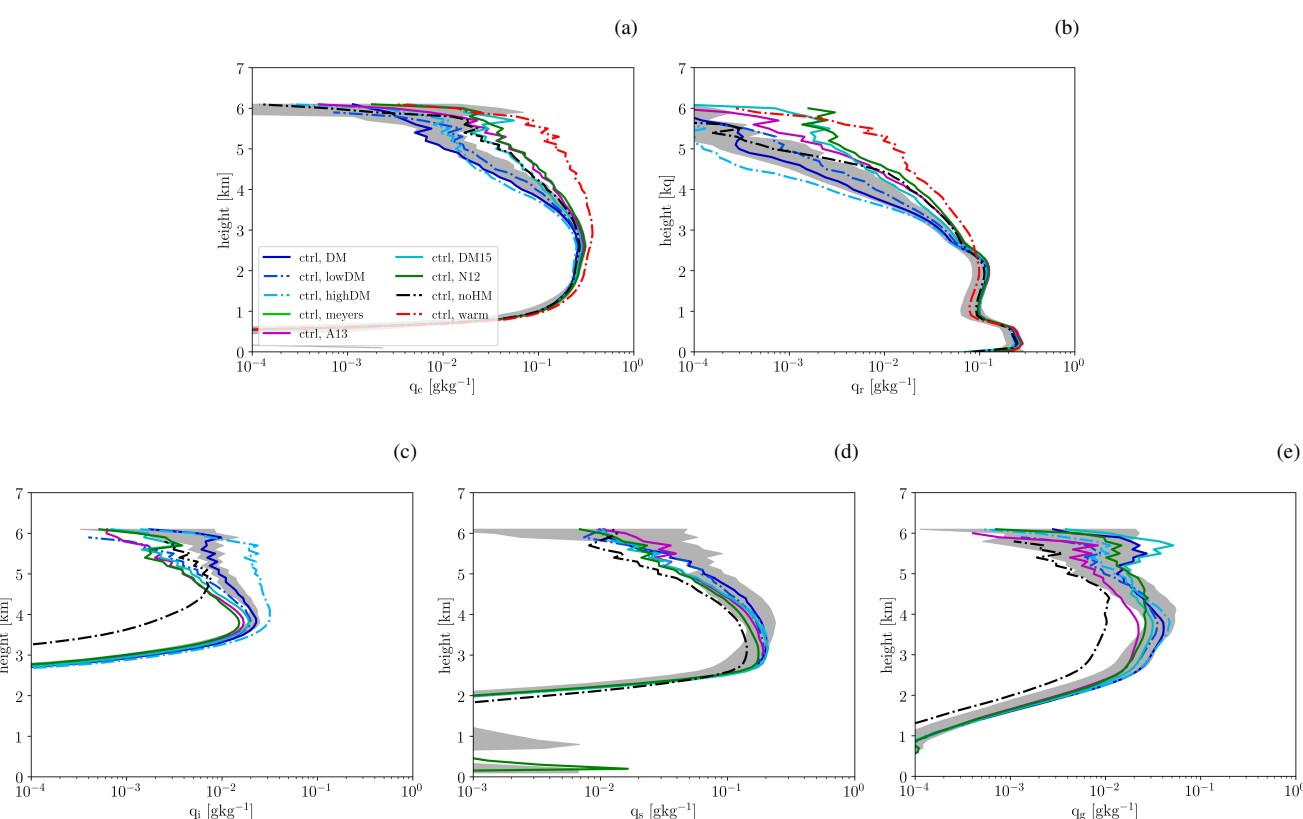

**Figure 2.** Same as Fig. 1 but for hydrometeor mass mixing ratios.

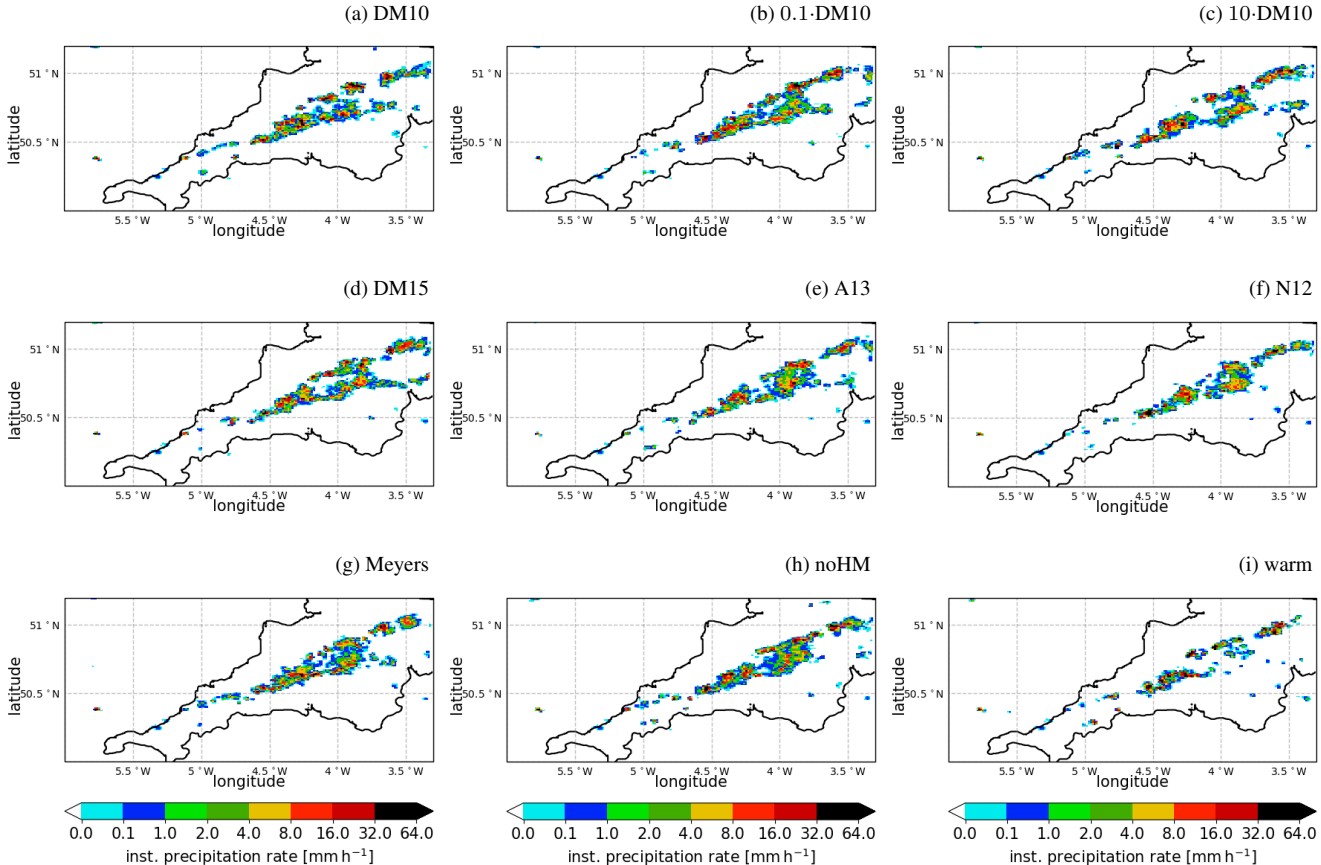

**Figure 3.** Instantaneous precipitation rate at 14 UTC in the different model simulations. (a), (b), and (c) show simulations using the DeMott et al. (2010) parameterisation, multiplied with factors 1, 0.1, and 10, respectively. The simulation in panel (a) corresponds to the "baseline" simulation. (d), (e), (f), and (g) show simulations using the parameterisations by DeMott et al. (2015), Atkinson et al. (2013), Niemand et al. (2012), and Meyers et al. (1992). (h) shows the simulation with the DeMott et al. (2010) parameterisation, but without a parameterisation of the Hallet-Mossop processes and, finally, (i) shows the simulation without any representation of ice formation.

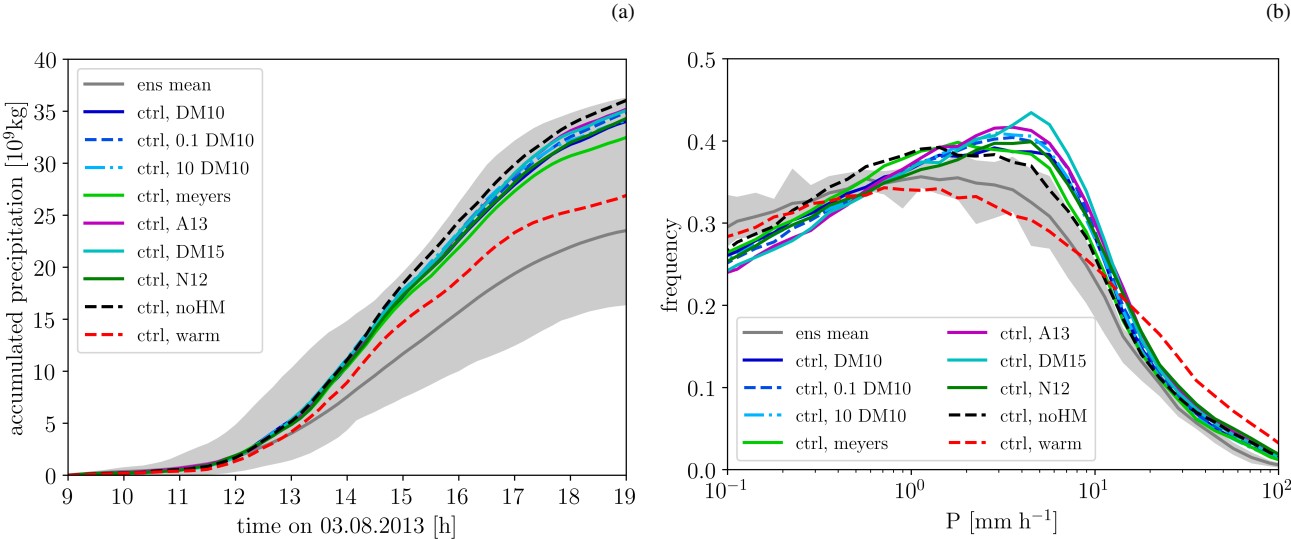

**Figure 4.** (a) Time series of accumulated surface precipitation. (b) Precipitation rate distribution (excluding non-raining grid-points). The dark grey shading shows the spread of the 10 ensemble members with perturbed initial conditions. The grey line represents the ensemble mean and the various coloured lines simulations with different heterogeneous freezing parameterisation, pure warm-phase microphysics and no Hallett-Mossop process (colours according to legend).

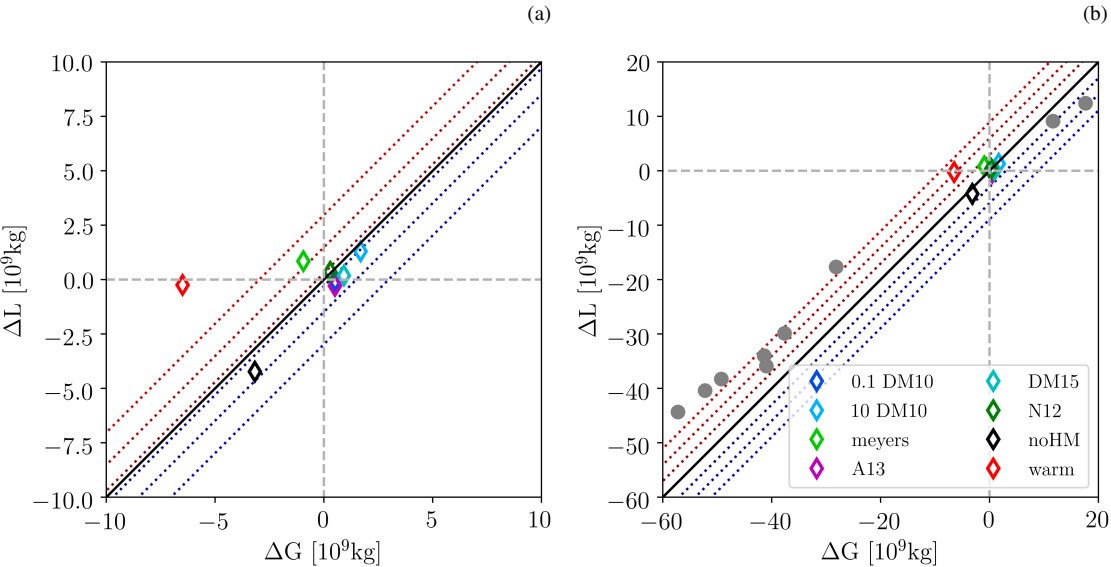

**Figure 5.** Scatterplot of change in condensate gain $\Delta G$ and condensate loss $\Delta L$ relative to the simulation with the DeMott et al. (2010) heterogeneous freezing parameterisation and a representation of the Hallett-Mossop process (baseline simulation). The condensate gain in the baseline simulation is $137.0\ 10^9$ kg and the condensate loss $107.3\ 10^9$ kg. The grey symbols in panel (b) represent the 9 meteorological ensemble members other than the baseline simulation. The blue and red dashed lines indicate relative changes in precipitation of 0.1, 5, 10 % in (a) and 10, 20, 30 % in (b).

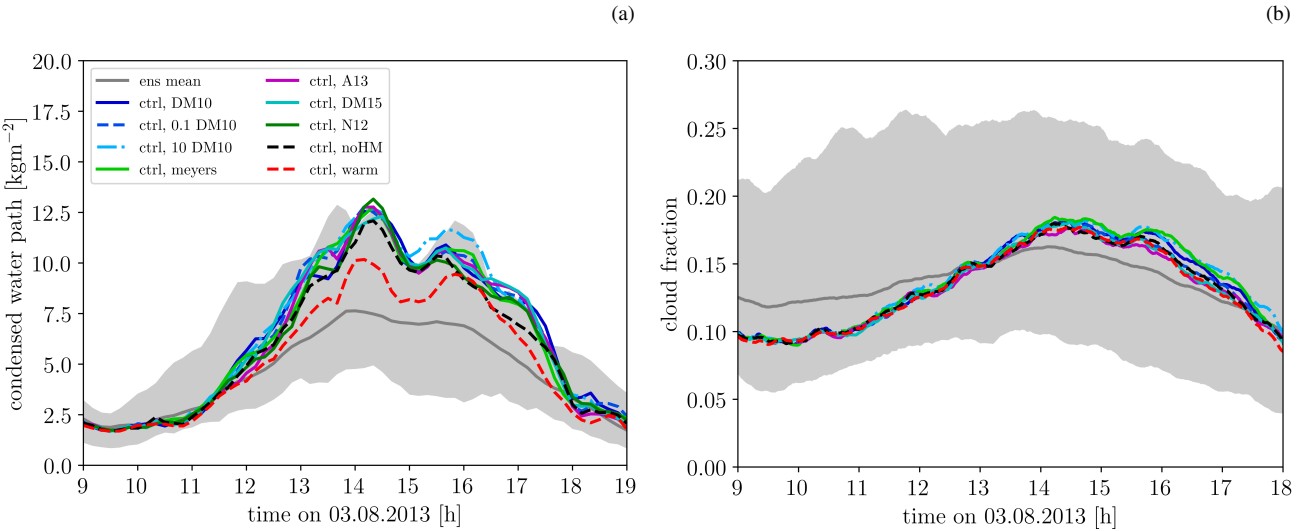

**Figure 6.** Time series of (a) the average condensed water path and (b) the cloud fraction. The dark grey shading in both panels shows the spread of the 10 ensemble members with perturbed initial conditions. The grey line represents the ensemble mean and the various coloured lines simulations with different heterogeneous freezing parameterisation, pure warm-phase microphysics and no Hallett-Mossop process (colours according to legend).

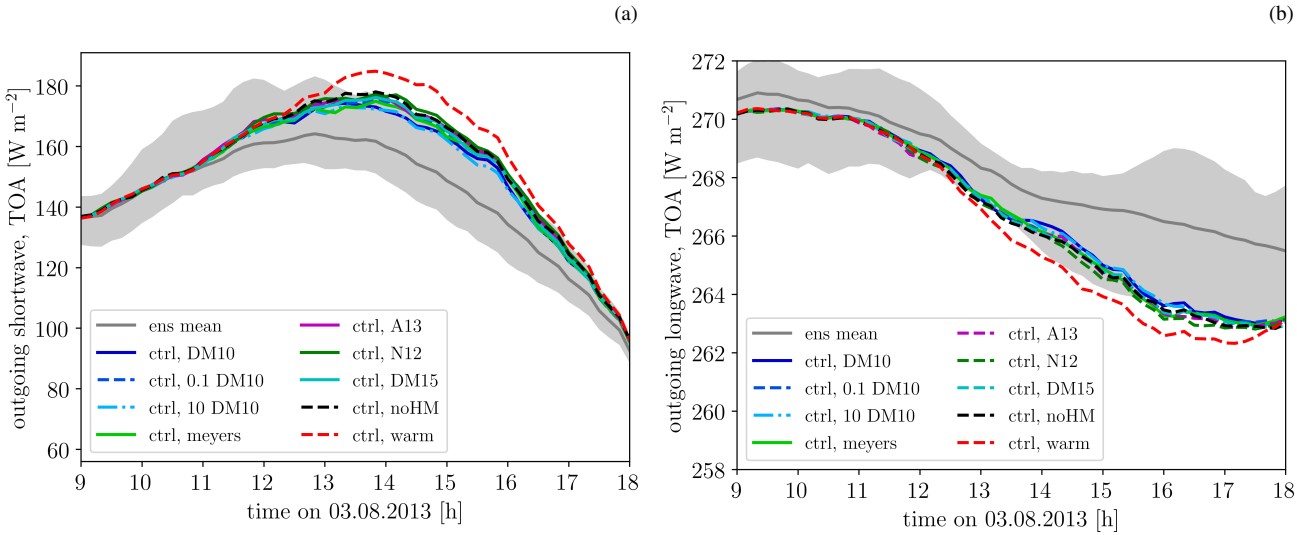

**Figure 7.** Domain-average time series of outgoing (a) shortwave and (b) longwave radiation at the top of atmosphere. The dark grey shading shows the spread of the 10 ensemble members with perturbed initial conditions. The grey line represents the ensemble mean and the various coloured lines simulations with different heterogeneous freezing parameterisation, pure warm-phase microphysics and no Hallett-Mossop process.

**Table A1.** Aerosol number concentration N, mass density m, and width of the size distribution $\sigma$ prescribed as lateral boundary and initial condition in the boundary layer.

|  | N $(\text{cm}^{-3})$ | m $(\text{kg m}^{-3})$ | $\sigma$ (1) |
|---|---|---|---|
| Aitken mode | 860 | $5.86 \cdot 10^{-10}$ | 2.2 |
| Accumulation mode | 150 | $3.84 \cdot 10^{-9}$ | 1.7 |
| Coarse mode | 0.23 | $1.07 \cdot 10^{-8}$ | 1.5 |
| Insoluble mode | 16.7 | $4.26 \cdot 10^{-10}$ | 1.5 |