# Peer review of "Sensitivity of mixed-phase moderately deep convective clouds to parameterisations of ice formation - An ensemble perspective"

_Atmospheric Chemistry and Physics, 2020_

## Referee Comment (RC1) · Anonymous Referee #1 · 6 Jul 2020

Recommendation: Major Revision

This work addresses the topic of sensitivity of fine-scale model predictions of convective clouds to different representations of ice formation and to initial conditions uncertainty. Mean hydrometeor profiles as well as bulk cloud field properties are examined for one convective case. The manuscript addresses a very interesting and up-to date topic. However, it suffers from the fact that it is in a way a "Part 3" of other manuscripts (Miltenberger et al., 2018a and b). The readability can be increased by adding information about the meteorological situation, the microphysical parameterisations and

their prediction on the tested case. Some explanations are missing. A damaging point is that we do not withdraw physical information on the validity of microphysical tests. The discussion could also be strengthened a bit on the possible generalization to other meteorological cases from this case study. Therefore, the manuscript requires major revision prior to acceptance for publication.

Major comments:

1. In general, this manuscript heavily depends on Miltenberger et al., 2018a and b of aerosol-cloud interactions in mixed-phase convective clouds. However, it should also be readable without the knowledge of the contents of Part 1 and 2. Therefore, I suggest that the authors give more information about: - the case study and the measurements used to validate the control simulation - the CASIM microphysical scheme : 2-moment scheme for all hydrometeor species, method of aerosol initialization - the physical configuration of the simulation: turbulent scheme, subgrid condensation scheme or not, radiation scheme I understand that it is not necessary to give all the information presented in the 1st paper but the reader should not be forced to seek all the information in the previous articles.

2. The seven heterogeneous freezing parameterisations introduced in this paper are not sufficiently presented. The main differences between DM10 (used as the control simulation) and M92, A13, DM15, N12 and T13 need to be explained.

3. It is said that DM10 successfully captures many features of the observed cloud and precipitation, but what about the other FSENS experiments? A figure like Fig.2 of Miltenberger et al. (2018b) applied to FSENS is necessary. It would be interesting for instance to see if NoHM modifies significantly the spatial pattern of clouds and surface precipitation.

4. Explanations about the cloud condensate budget used in Khain (2009) and Miltenberger et al. (2018a) presented in Fig.4 are missing.

[Figure]

5. In the discussion, the authors have anticipated the question whether this dominance of initial condition uncertainty is a special feature of the chosen case. We understand that it is difficult to answer on the basis of this study. But they may try to discuss about the extension of the analysis to other cloud types and meteorological scenarios.

No minor comment : The text is well written and figures are clear.

---

## Referee Comment (RC2) · Anonymous Referee #3 · 29 Dec 2020

The paper addresses an issue of interest, and I believe it deserves to reach out the scientific community working on modelling of ice formation in clouds.

Overall, I share the same doubts and reserves as the other reviewer as long as the readability of the current paper is concerned. The authors need to make the paper self consistent and easier to read. Please explain all the assumptions taken and describe the parameterizations with some level of details. Just referencing to existing works is not enough! I'm not familiar with the work of Miltenberger et al., don't expect that other readers will be.

[Figure]

Please consider adding a table showing the main features of each parameterization, guiding the reader through your methodology. The commonalities and differences of each scheme is functional to discuss the spread of the ensemble. Without providing info about the diversity/commonality of the underlying assumptions of each scheme, how is possible to interpret if the spread of the ensemble reflects true physical uncertainty? Perhaps all schemes descend from the same physical assumptions, in that case I would expect an overconfident ensemble spread. As the paper stands at this stage, it cannot be deduced.

Another obscure point to me is the use (or not use) of 'observational data'. At the beginning of section 2 the COPE campaign is mentioned. What about using the data collected there to shed some light on the bias/error of the modelling results? if this is part of the baseline simulation it needs to be clarified. Maybe I'm missing something, but I believe that the use of measurements could enormously add value to the current findings (at least, if possible, for one variable; I believe it'd be very important if you did).

On a less general note:

- consider adding a description of Hallet-Mossop process (and maybe acronym it to H-M);

- consider give percentage of the values in table 1, absolute magnitude alone doesn't say much about variability;

- line 8: perhaps you meant 'changes' rather than 'change'?

---

## Author Comment (AC1) · 5 Feb 2021

**Reply to Anonymous Referee #1**

This work addresses the topic of sensitivity of fine-scale model predictions of convective clouds to different representations of ice formation and to initial conditions uncertainty. Mean hydrometeor profiles as well as bulk cloud field properties are examined for one convective case. The manuscript addresses a very interesting and up-to date topic. However, it suffers from the fact that it is in a way a "Part 3" of other manuscripts (Miltenberger et al., 2018a and b). The readability can be increased by adding information about the meteorological situation, the microphysical parameterisations and their prediction on the tested case. Some explanations are missing. A damaging point is that we do not withdraw physical information on the validity of microphysical tests. The discussion could also be strengthened a bit on the possible generalization to other meteorological cases from this case study. Therefore, the manuscript requires major revision prior to acceptance for publication.

**Major comments:**

- 1. In general, this manuscript heavily depends on Miltenberger et al., 2018a and b of aerosol-cloud interactions in mixed-phase convective clouds. However, it should also be readable without the knowledge of the contents of Part 1 and 2. Therefore, I suggest that the authors give more information about:
  - the case study and the measurements used to validate the control simulation
  - the CASIM microphysical scheme : 2-moment scheme for all hydrometeor species, method of aerosol initialization
  - the physical configuration of the simulation: turbulent scheme, subgrid condensation scheme or not, radiation scheme.

I understand that it is not necessary to give all the information presented in the 1st paper but the reader should not be forced to seek all the information in the previous articles.

**Reply:** The requested information has been added to the manuscript. Some details on CASIM and the aerosol initial / lateral boundary conditions were added to the main text (l. 74-75 & 80-84 of the new manuscript). Other information is included in new Appendix A for the model set-up.

More details on the comparison to observations for the baseline simulations are summarised in I. 120-148 of the new manuscript.

2. The seven heterogeneous freezing parameterisations introduced in this paper are not sufficiently presented. The main differences between DM10 (used as the control simulation) and M92, A13, DM15, N12 and T13 need to be explained.

**Reply:** We have added a more detailed description of the different parameterisations and the resulting difference in the temperature-dependence of INP number concentration, which is the main impact of the different parameterisations (I. 99-116 of the new manuscript).

3. It is said that DM10 successfully captures many features of the observed cloud and precipitation, but what about the other FSENS experiments? A figure like Fig. 2 of Miltenberger et al. (2018b) applied to FSENS is necessary. It would be interesting for instance to see if NoHM modifies significantly the spatial pattern of clouds and surface precipitation.

**Reply:** A figure similar to Fig. 2 of Miltenberger et al. (2018b) was included in the paper showing the instantaneous precipitation rate at 14 UTC, i.e. a time close to the maximum precipitation rate of the day (Fig. 3 of the new manuscript). Structurally the precipitation fields show only minor differences, with the exception of the "WARM" simulation, that are comparable to the differences between different initial condition ensembles. In the "WARM" simulation a clear shift to more localised and more intense precipitation rates is observed. The figures and its implications for the study are discussed in I. 186-191 of the new manuscript.

4. Explanations about the cloud condensate budget used in Khain (2009) and Miltenberger et al. (2018a) presented in Fig.4 are missing.

**Reply:** We have included a few sentences introducing the cloud condensate budget analysis and some additional explanation on the figure (I. 197-211).

5. In the discussion, the authors have anticipated the question whether this dominance of initial condition uncertainty is a special feature of the chosen case. We understand that it is difficult to answer on the basis of this study. But they may try to discuss about the extension of the analysis to other cloud types and meteorological scenarios.

**Reply:** We appreciate the interest of the reviewer with regard to the extrapolation of the results for the investigated case towards other scenarios. As stated by the reviewer such an extrapolation is difficult, but we have nevertheless added a few comments and thoughts towards the end of the discussion section (I. 362-373 of the new manuscript).

Here, we would like to also point out that there is work going on at the moment with regard to the extension to a larger variety of cases, but it is too early to comment in detail on these studies.

No minor comment : The text is well written and figures are clear.

Reply: Thank you!

---

## Author Comment (AC2) · 5 Feb 2021

**Reply to Anonymous Referee #3**

*The paper addresses an issue of interest, and I believe it deserves to reach out the scientific community working on modelling of ice formation in clouds.*
*Overall, I share the same doubts and reserves as the other reviewer as long as the readability of the current paper is concerned. The authors need to make the paper self consistent and easier to read. Please explain all the assumptions taken and describe the parameterizations with some level of details. Just referencing to existing works is not enough! I'm not familiar with the work of Miltenberger et al., don't expect that other readers will be.*

**Reply:** We have added more detail on the set-up of the simulations (Appendix A, l. 74-75, and l. 80-84 of the new manuscript) and also the comparison to observations as reported in our earlier work on this case (l. 120-148).

*Please consider adding a table showing the main features of each parameterization, guiding the reader through your methodology. The commonalities and differences of each scheme is functional to discuss the spread of the ensemble. Without providing info about the diversity/commonality of the underlying assumptions of each scheme, how is possible to interpret if the spread of the ensemble reflects true physical uncertainty? Perhaps all schemes descend from the same physical assumptions, in that case I would expect an overconfident ensemble spread. As the paper stands at this stage, it cannot be deduced.*

**Reply:** We have added a more detailed description of the different parameterisations and the resulting difference in the temperature-dependence of INP number concentration, which is the main impact of the different parameterisations (l. 99-116 of the new manuscript).

*Another obscure point to me is the use (or not use) of 'observational data'. At the beginning of section 2 the COPE campaign is mentioned. What about using the data collected there to shed some light on the bias/error of the modelling results? if this is part of the baseline simulation it needs to be clarified. Maybe I'm missing something, but I believe that the use of measurements could enormously add value to the current findings (at least, if possible, for one variable; I believe it'd be very important if you did).*

**Reply:** A paragraph summarising the comparison with observational data for the baseline case in Miltenberger et al (2018a) has been included (l. 120-148 of the new manuscript). Here we also detail how the comparison to radiosonde and radar data differ for the new sensitivity experiments. We agree that a more detailed comparison to for example in-situ observations of ice-crystal number density or 3D radar reflectivity structure would be interesting. However, the model data is either not available (radar reflectivity) or is not available at a sufficiently high time resolution to allow for a meaningful comparison (in-situ data is mainly sampling rising cloud tops). We briefly comment on this in the abstract.

*On a less general note:*
- *consider adding a description of Hallet-Mossop process (and maybe acronym it to H-M);*

    **Reply:** We have added a short description of the Hallet-Mossop processes in l. 93-96 of the new manuscript.

- *consider give percentage of the values in table 1, absolute magnitude alone doesn't say much about variability;*

    **Reply:** We have altered the table to the relative spread in the variables, i.e. the ratio between the difference between maximum and minimum value to the value in the baseline simulation (Tab. 1 of the new manuscript).

- *line 8: perhaps you meant 'changes' rather than 'change'?*

    **Reply:** Thanks for spotting this, this is corrected in the new version.